# Comparative transcriptional profiling of the early host response to infection by typhoidal and non-typhoidal *Salmonella* serovars in human intestinal organoids

Basel H. Abuaita[1☉], Anna-Lisa E. Lawrence[1☉], Ryan P. Berger[1], David R. Hill[2], Sha Huang[3], Veda K. Yadagiri[2], Brooke Bons[2], Courtney Fields[2], Christiane E. Wobus[1], Jason R. Spence[3], Vincent B. Young[2], Mary X. O'Riordan[1]*

1 Department of Microbiology and Immunology, University of Michigan Medical School, Ann Arbor, Michigan, United States of America, 2 Department of Internal Medicine, University of Michigan Medical School, Ann Arbor, Michigan, United States of America, 3 Department of Cell and Developmental Biology, University of Michigan Medical School, Ann Arbor, Michigan, United States of America

☉ These authors contributed equally to this work.
* oriordan@umich.edu

**Data Availability Statement:** All sequences are deposited in the EMBL-EBI Arrayexpress database (E-MTAB-10451). Source code for analyses can be

## Abstract

*Salmonella enterica* represents over 2500 serovars associated with a wide-ranging spectrum of disease; from self-limiting gastroenteritis to invasive infections caused by non-typhoidal serovars (NTS) and typhoidal serovars, respectively. Host factors strongly influence infection outcome as malnourished or immunocompromised individuals can develop invasive infections from NTS, however, comparative analyses of serovar-specific host responses have been constrained by reliance on limited model systems. Here we used human intestinal organoids (HIOs), a three-dimensional "gut-like" *in vitro* system derived from human embryonic stem cells, to elucidate similarities and differences in host responses to NTS and typhoidal serovars. HIOs discriminated between the two most prevalent NTS, *Salmonella enterica* serovar Typhimurium (STM) and *Salmonella enterica* serovar Enteritidis (SE), and typhoidal serovar *Salmonella enterica* serovar Typhi (ST) in epithelial cell invasion, replication and transcriptional responses. Pro-inflammatory signaling and cytokine output was reduced in ST-infected HIOs compared to NTS infections, consistent with early stages of NTS and typhoidal diseases. While we predicted that ST would induce a distinct transcriptional profile from the NTS strains, more nuanced expression profiles emerged. Notably, pathways involved in cell cycle, metabolism and mitochondrial functions were downregulated in STM-infected HIOs and upregulated in SE-infected HIOs. These results correlated with suppression of cellular proliferation and induction of host cell death in STM-infected HIOs and in contrast, elevated levels of reactive oxygen species production in SE-infected HIOs. Collectively, these results suggest that the HIO model is well suited to reveal host transcriptional programming specific to infection by individual *Salmonella* serovars, and that individual NTS may provoke unique host epithelial responses during intestinal stages of infection.

found at: https://github.com/rberger997/HIO_dualseq2 and https://github.com/aelawren/Salmonella-serovars-RNA-seq.

**Funding:** This work was supported by NIH NIAID U19AI116482 (JRS, VBY, CEW, MXO). A-LEL was supported by the Molecular Mechanisms of Microbial Pathogenesis training grant (NIH T32 AI007528). The funders had no role in study design, data collection and analysis, decision to publish, or preparation of the manuscript.

**Competing interests:** The authors have declared that no competing interests exist.

## Author summary

*Salmonella enterica* is the major causative agent of bacterial infections associated with contaminated food and water. *Salmonella enterica* consists of over 2500 serovars of which Typhimurium (STM), Enteritidis (SE) and Typhi (ST) are the three major serovars with medical relevance to humans. These serovars elicit distinctive immune responses and cause different diseases in humans, including self-limiting diarrhea, gastroenteritis and typhoid fever. Differences in the human host response to these serovars are likely to be a major contributing factor to distinct disease outcomes but are not well characterized, possibly due to the limitations of human-derived physiological infection models. Distinct from immortalized epithelial cell culture models, human intestinal organoids (HIOs) are three-dimensional structures derived from embryonic stem cells that differentiate into intestinal mesenchymal and epithelial cells, mirroring key organizational aspects of the intestine. In this study, we used HIOs to monitor transcriptional changes during early stages of STM, SE and ST infection. Our comparative analysis showed that HIO inflammatory responses are the dominant response in all infections, but ST infection induces the weakest upregulation of inflammatory mediators relative to the other serovars. In addition, we identified several cellular processes, including cell cycle and mitochondrial functions, that were inversely regulated between STM and SE infection despite these serovars causing similar localized intestinal infection in humans. Our findings reinforce HIOs as an emerging model system to study *Salmonella* serovar infection and define global host transcriptional response profiles as a foundation for understanding human infection outcomes.

## Introduction

*Salmonella enterica* greatly impacts human health causing an estimated 115 million infections worldwide every year and are one of the four leading causes of diarrheal diseases [1,2]. *Salmonella enterica* consists of over 2500 serovars and infects the intestinal epithelial layer, causing a wide spectrum of phenotypes ranging from asymptomatic carriage to more severe systemic disease. *Salmonella* serovars are classified based on host specificity and disease outcomes. Host generalist serovars including *Salmonella enterica* serovar Typhimurium (STM) and Enteritidis (SE) infect a broad range of hosts and cause localized inflammation and self-limiting diarrhea in healthy individuals or more severe gastroenteritis in children and the elderly. In contrast, host-restricted serovars including Typhi and Abortusovis infect only one host species and cause more serious clinical manifestations including typhoid fever in humans and abortions in mares respectively [3].

Although *Salmonella enterica* serovars share a conserved core genome, determinants of host specificity and varying clinical manifestations are poorly understood. The molecular basis for distinct host adaptation and disease outcome is likely to be multifactorial, mediated by bacteria and host-dependent mechanisms [4]. Initial comparative genomic analyses identified specific signatures that may be indicative of some of these differences [5–7]. However, comparison of host signatures across different serovars is still limited by host specificity and poorly representative model systems. Using human epithelial cell lines addresses host-specificity, but immortalized cell lines do not represent the multiple subsets of intestinal epithelial cells found in the gut and harbor mutations that likely alter cellular responses to bacterial infection.

Human intestinal organoids (HIOs) have emerged as an alternative *in vitro* model to study intestinal epithelial host responses to commensal microbiota and enteric pathogens [8]. HIOs are differentiated from pluripotent stem cells into three-dimensional spheroids composed of a defined luminal space bound by a polarized epithelial barrier surrounded by mesenchyme. This is an improvement over existing models because the untransformed HIO epithelium is polarized and contains multiple epithelial cell lineages found in the intestine [9]. Hill *et al.* showed that HIOs supported luminal growth of *Escherichia coli* following microinjection, and that physiological changes in the HIO occurred during colonization, such as an increase in mucus production, mirroring what happens *in vivo* during initial colonization [10]. Our work and Forbester *et al.* also showed that STM invades HIO epithelial cells and induces inflammatory responses, suggesting that the HIO is an effective model to define intestinal host responses to enteric pathogens [11,12]. Here, we used HIOs to compare the transcriptional profiles of intestinal epithelial responses to host-restricted *Salmonella enterica* serovar Typhi and two host unrestricted serovars Typhimurium and Enteritidis. We found that *Salmonella* infection induced a variation in magnitude of immune responses that was dependent on the infecting serovar. ST infection induced the weakest response, consistent with the idea that ST infection induces a weak host immune response to establish a systemic infection [4]. Notably, we found that both STM and ST infection similarly decreased expression of pathways involved in cell cycle, DNA repair and DNA replication while SE infection increased these responses.

## Results

### *Salmonella* serovars invade HIO epithelial cells and induce distinct patterns of mucus production

To study initial host responses to *Salmonella*, we microinjected bacteria into the luminal space of the HIO to allow luminal replication throughout the course of infection (**Fig 1A**). This HIO infection model allows for longer-term interactions between bacteria and the host both in the extracellular luminal space and intracellularly within epithelial cells and therefore, it better resembles the continuous interaction between bacteria and intestinal cells during the natural course of infection. We first determined whether different *Salmonella* serovars could colonize and replicate within the HIOs and invade HIO epithelial cells. We chose the most prevalent serovars that cause gastroenteritis, STM and SE, and a typhoidal serovar, ST. HIOs were microinjected with $10^3$ CFU of STM, SE or ST, a relatively low inoculum, as previous work demonstrated that growth rate was negatively correlated with the number of CFU injected [10]. Total bacterial burden per HIO was enumerated at 2.5 hours post-infection (hpi) to establish initial levels of colonization, and 24 hpi (**Fig 1B**). All serovars showed at least a 1.5 log increase in bacterial burden at 24hpi, relative to 2.5hpi. Intracellular bacterial burden was quantified by gentamicin protection assay, as we previously showed the utility of this assay in the HIOs by comparing invasion between WT STM and a *Salmonella* pathogenicity island-1 (SPI-1) deficient isogenic mutant [12]. Briefly, HIOs were cut open to expose luminal bacteria to gentamicin before lysing epithelial cells for enumeration of gentamicin-protected bacteria (**Fig 1C**). Intracellular bacteria numbers increased over time with all three serovars, suggesting that intracellular replication or continued invasion contributes to increased bacterial load at 24hpi. At this low inoculum, STM consistently invaded HIO epithelial cells more efficiently than SE and ST. To determine whether continued invasion or intracellular replication explained the increase in intracellular burden over time, infected HIOs were sectioned and stained with DAPI and anti-E-Cadherin antibody to visualize DNA and epithelial cells respectively, allowing quantification of the number of bacteria per cell (**Fig 1D–1F**). There were more bacteria per cell (**Fig 1D**) and a higher percentage of infected cells (**Fig 1F**) in STM-

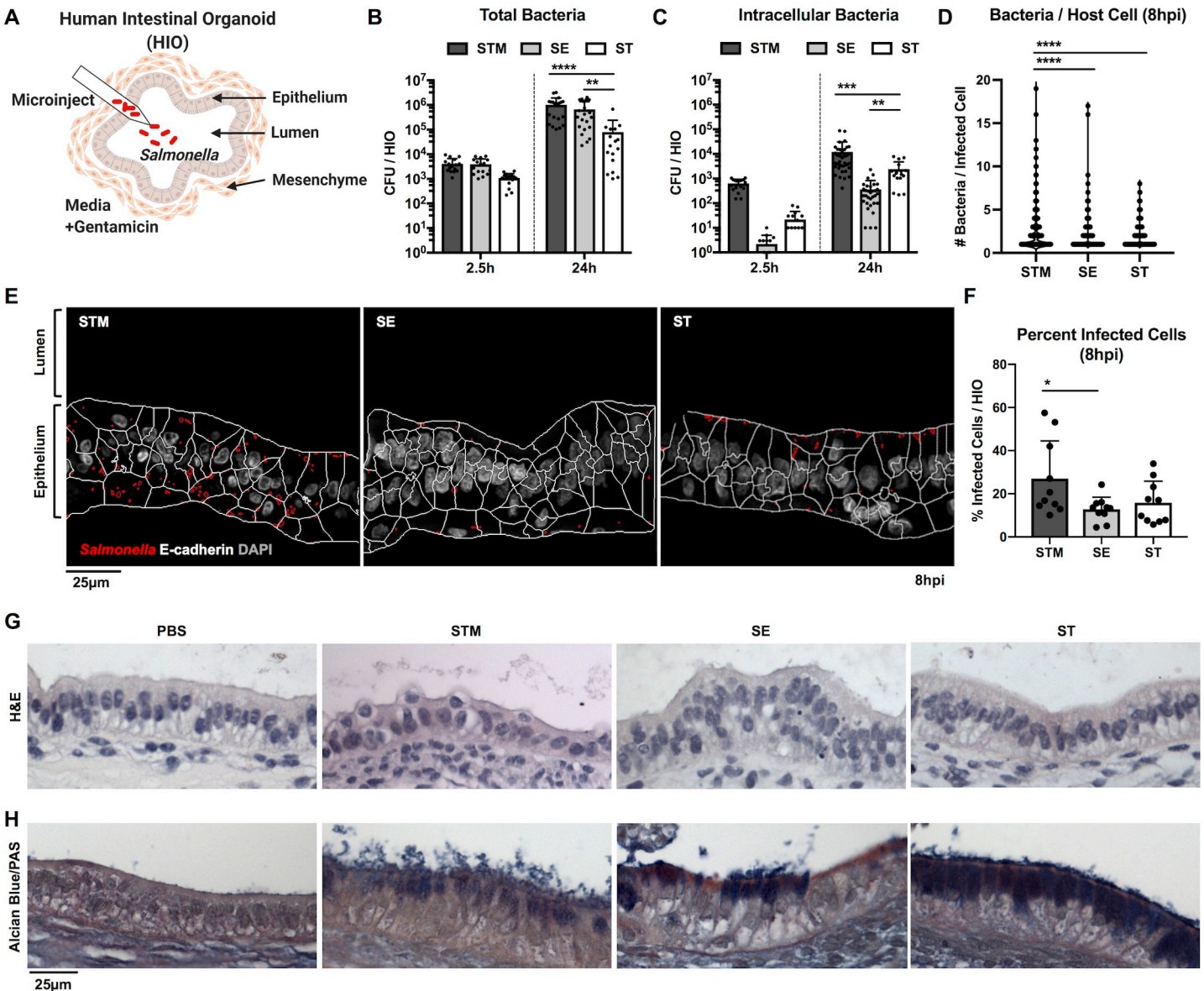

**Fig 1. *Salmonella enterica* serovars invade HIO epithelial cells and stimulate mucin production.** (A) Model depicting experimental set-up with the HIOs. HIOs are comprised of an epithelium lining surrounded by mesenchymal cells that self-organize into 3-dimensional structures. Bacteria ($10^3$ CFU) were microinjected into the HIO lumen and gentamicin was added to the medium after 2h to kill any bacteria introduced into the medium during microinjection. (B) Total bacterial burden was enumerated per HIO at 2.5h and 24h post infection. (C) Intracellular bacterial burden was enumerated after exposing luminal bacteria to gentamicin at 2.5h and 24h post infection. Graphs represent the mean of n>16 HIOs from three independent experiments. Statistical significance within the group was determined by two-way ANOVA and followed up by Tukey's multiple comparisons test. (D) Number of bacteria per cell was quantified using DAPI staining. (E) Representative confocal microscopy images of histology sections obtained from STM, SE, or ST infected HIOs for 8h. Sections were stained for E-cadherin and DAPI. Cell outlines based on the E-cadherin staining (white) and bacterial outlines detected using DAPI staining (red) were generated using CellProfiler. (F) Percent of infected cells were determined by quantifying 3 fields of view per HIO at 60x magnification with n = 10 HIOs analyzed per group. (G and H) Histology sections of HIOs at 8hpi using hematoxylin and eosin (H&E) staining (G) and Alcian Blue/Periodic Acid-Schiff (PAS) staining (H). Statistical significance for (D) and (F) was determined by one-way ANOVA with Tukey's multiple comparisons test. P values < 0.05 were considered significant and designated by: *<0.05, **<0.01, ***<0.001 and ****<0.0001.

infected HIOs compared to SE- or ST-infected HIOs, supporting the conclusion that STM invades more efficiently than the other two serovars when initial bacterial numbers are low. Most infected cells contained 1–2 bacteria, which might reflect continued invasion over time. However, particularly in STM-infected HIOs, where we measured a marked increase in CFU,

we observed some cells that contained >10 bacteria per cell consistent with intracellular replication of *Salmonella*. Because there were differences in intracellular bacterial burden between serovars, we compared the expression of SPI-1 and SPI-2 genes during infection as differences in virulence gene expression may contribute to differences in invasion and intracellular replication. Transcriptional analysis of *Salmonella* genes from infected HIOs revealed that expression of SPI-1 genes was highest in STM at 2.5hpi, suggesting that enhanced expression of effectors mediating invasion, such as SopB or SopE may allow STM to enter HIO cells more efficiently than the other two serovars (**S1 Fig**). Expression of SPI-1 effector genes decreased in all serovars over time while SPI-2 effector expression increased, indicating that the HIO environment reprograms bacterial gene expression. In order to determine the impact of maintaining live bacteria in the HIO lumen throughout the course of infection on HIO integrity and morphology, we performed hematoxylin and eosin (H&E) histology staining (**Fig 1G**). HIOs remained intact during infection with all serovars for the duration of the experiment. However, stressed regions of the HIO epithelial lining could sometimes be observed, especially during STM infection, where epithelial cells appeared to be extruded into the lumen. In addition, Alcian Blue and Periodic acid-Schiff reagent (PAS) staining was also performed to detect mucus, as a recent study showed that HIOs increase mucin production during bacterial colonization [10]. In agreement with these findings, Alcian Blue and PAS staining showed an increase in mucus production in response to infection (**Figs 1H and S2**). Of note, we observed unique staining patterns during infection with the different serovars. While STM infection resulted in luminal mucus accumulation, in ST-infected HIOs, mucus accumulated within epithelial cells, indicating that serovars can differentially modulate mucus production or secretion. Taken together, our data show that over a 24h period, all three *Salmonella* serovars colonize HIOs, and invade HIOs, inducing distinct patterns of mucus production without causing major destruction to the HIO epithelial layer.

## Host transcriptional dynamics differ between *Salmonella* serovars

To define the global HIO transcriptional response to the three *Salmonella* serovars, we performed RNA sequencing (RNA-seq) at 2.5h and 8hpi. HIOs were infected with $10^5$ CFU of STM, SE or ST. A higher inoculum was used in order to establish comparable bacterial loads in the HIOs at 8h, and transcriptional changes were compared relative to control PBS-injected HIOs (**S3 Fig**). Principal component analysis (PCA) was performed on normalized gene counts to identify clustering patterns between conditions (**Fig 2A–2C**). PCA plots showed clear segregation and clustering of samples based on both infection and time. Infected HIOs at 2.5h had the most variation relative to PBS where they were separated by the first (the greatest variance) principal component and further clustered based on infection with each serovar. STM-infected HIOs showed the greatest separation from the control, while SE-infected HIOs showed an intermediate and ST-infected HIOs showed the least separation. Infected HIOs at 8h were further separated by both the first and second (the second greatest variance) principal components. By 8hpi, the variation observed through the first component was decreased relative to 2.5h, suggesting that some early responses were transient (**Fig 2B and 2C**). At 8h, there was less segregation between infected HIOs and PBS, with no clear clustering of serovars at this later time point.

To further define HIO responses during *Salmonella* infection, we identified differentially expressed genes (DEGs) between PBS controls and HIOs infected with STM, SE or ST (**S1 and S2 Tables**). We found comparable numbers of DEGs during infection with all serovars at 2.5hpi (**Fig 2D**). Some of the DEGs were shared between all infected HIOs, which likely represents a core host response to *Salmonella* infection. However, infection with each serovar also

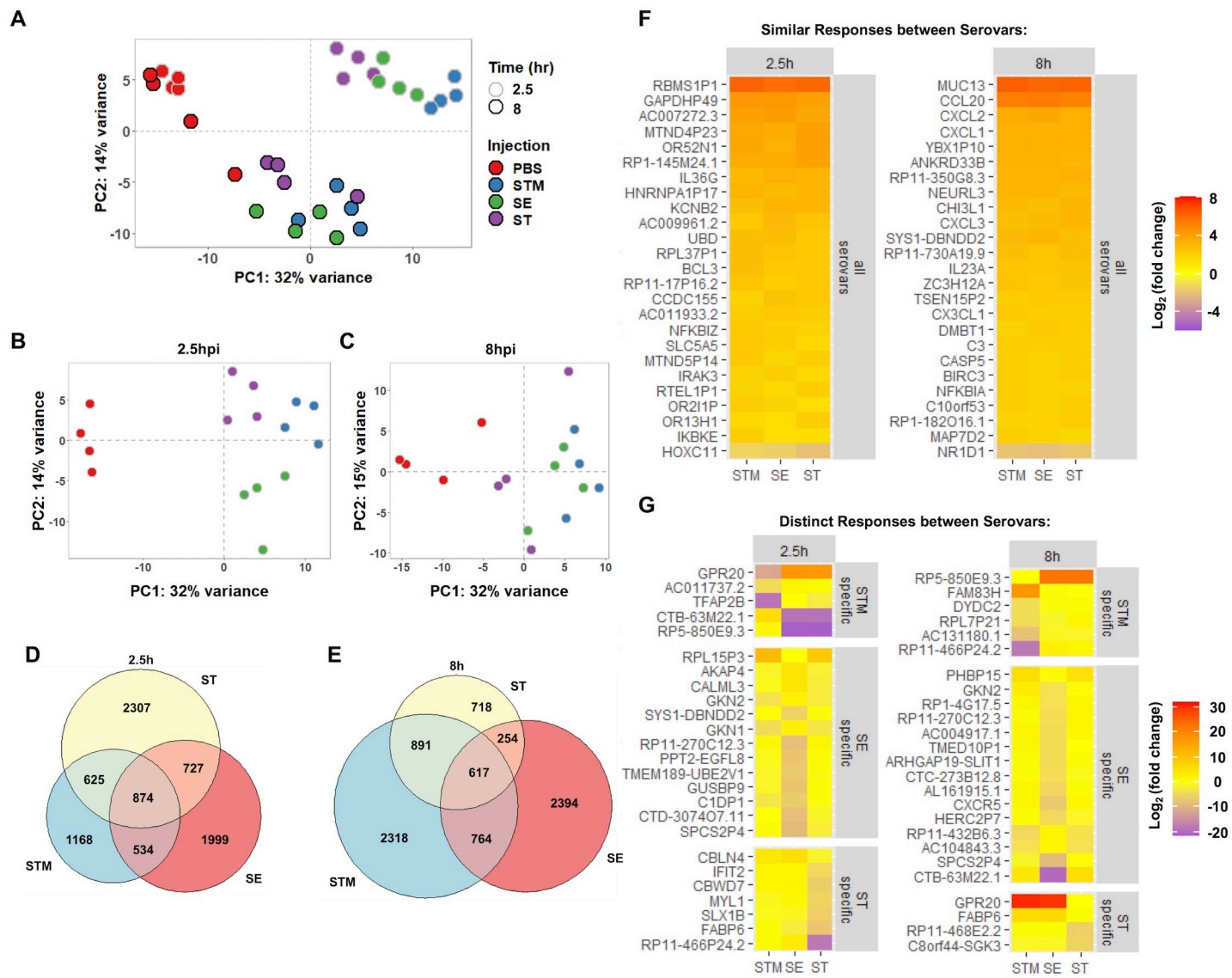

**Fig 2. Changes in HIO gene expression are driven by both serovar and time post infection.** (A) Principal component analysis of HIOs microinjected with $10^5$ CFU of the indicated *Salmonella* serovars. Each circle represents a biological replicate of a pool of five HIOs. (B and C) Principal component analysis of each time point 2.5h (B) and 8h (C). (D and E) Euler diagram comparison of gene expression changes in each condition relative to PBS at 2.5h (D) and 8h (E) post infection. Genes were filtered by P value < 0.05. (F and G) Heatmaps depicting conserved responses between serovars (F) and distinct responses between serovars (G) based on significant genes sorted by greatest standard deviation between conditions of $\log_2$(fold change) compared to PBS controls.

resulted in induction and suppression of a unique set of DEGs. We compared the DEGs from the HIOs with previously published *Salmonella* infection transcriptomics studies [11,13,14] and found that correlation between our dataset and the top responses to either STM or ST reported in each publication varied depending on the model system used (**S4 Fig**). There were high similarities in significant genes between our dataset and the dataset from the Forbester *et al.* study [11], in which a similar HIO model and wildtype STM strain was used. Notably, transcriptional dynamics from our analysis showed an increase in the number of DEGs at 8hpi in response to infection with the non-typhoidal serovars (NTS), STM and SE, while the number of DEGs during infection with ST decreased (**Fig 2E**). To better understand the conserved

and unique responses between serovars in the HIOs, significant DEGs were sorted by the standard deviation of the log₂(fold change) for each serovar and top 25 conserved genes between serovars and top 25 variable genes at each time point were plotted (**Fig 2F and 2G**). Included in the core response to all three serovars were proinflammatory mediators including upregulation of cytokine and chemokines (CCL20, CXCL2, CXCL1, CXCL3, IL23A, and IL36G), upregulation of components of the NF-κB signaling pathway (NFKBIZ, IKBKE, and NFKBIA) and one mucin (MUC13), among others. In contrast, the genes that comprised distinct responses to each serovar were involved in more diverse roles in the cell, including strong upregulation of the constitutively active G protein-coupled receptor (GPR20) in NTS [15] at 8 hpi, but not in ST-infected HIOs, and suppression of the intestinal fatty acid binding protein (FABP6) in ST-infected HIOs. Collectively, the HIO responses represent two patterns; core transcriptional responses that are changed during infection with all three *Salmonella* serovars and serovar-specific responses.

## *Salmonella* serovars differentially alter inflammatory, stress response, vesicular trafficking, metabolism and cell cycle pathways

To identify biological pathways associated with DEGs from each infection condition, gene sets were separated into upregulated (increased) and downregulated (decreased) categories based on fold change relative to PBS and imported separately into the Reactome pathway analysis tool (**S3–S6 Tables**). In the upregulated datasets, the majority of significant pathways in all three infection conditions at both 2.5h and 8h belonged to the immune system category with over 80 pathways significantly enriched in STM and SE-infected HIOs at 8hpi accounting for almost 5% of all annotated immune system pathways in the Reactome database (**Fig 3A**). We found that infection induced a complex response in both innate immune and cytokine

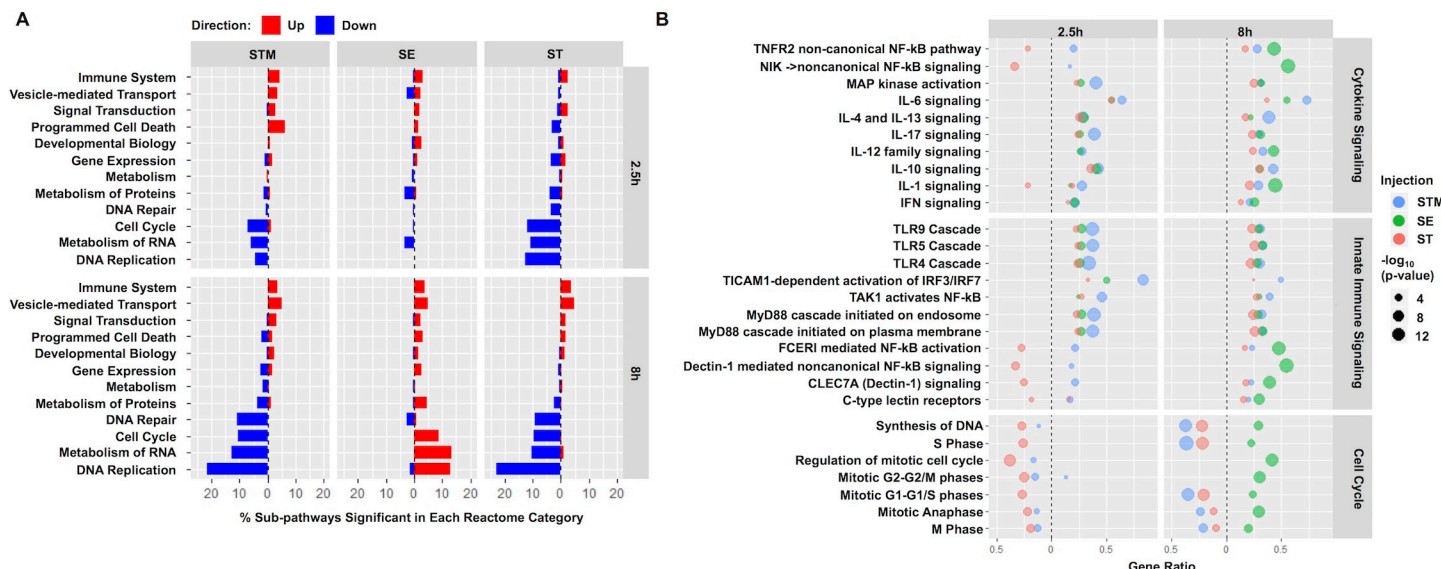

**Fig 3. Immune system and cell cycle pathways encompass the predominant increases and decreases in gene expression during infection.** (A) Fraction of sub-pathways clustering into the major Reactome cellular processes. Significantly upregulated (red) or downregulated (blue) genes were analyzed using ReactomePA and pathways were clustered into major cellular processes from the Reactome database. Major cellular processes with at least 12 significant sub-pathways in at least one infection condition were included with the proportion of significant pathways out of the entire group plotted on the x-axis. (B) Dot plot showing top pathways enriched from the Reactome database. Pathway coverage shown as gene ratio with significantly upregulated pathways shown on the right of the dotted line and downregulated pathways on the left. Dot size represents -log₁₀(p-value) of enriched pathway during HIO infection with STM in blue, SE in green and ST in red. Significant pathways were determined based on P value < 0.05.

signaling pathways including, but not limited to, Toll-like receptors, Interleukin mediators and Type I interferons (**Fig 3B**). Notably, only in ST-infected HIOs were some immune system pathways associated with downregulated DEGs, such as non-canonical NF-κB and Interleukin-1 signaling. These results revealed that inflammatory pathways were the primary responses during *Salmonella* infection and are consistent with the hypothesis that typhoidal serovar infection is relatively "silent", producing less inflammatory mediators compared to NTS infection.

Apart from the predominant inflammatory pathways, we also identified several differentially upregulated pathways in response to *Salmonella* serovars that have been linked to intestinal infection. These pathways included vesicular mediated transport, antigen presentation, extracellular matrix organization (ECM), lipid and amino acid metabolism and cellular stress responses including IRE1α-mediated unfolded protein response (UPR), mitophagy and the inflammasome (**S5 Fig**). Although there were genes in these pathways that were significantly upregulated in response to all three serovars, some were enriched only in response to a specific serovar. For example, we found that pathways belonging to ECM, UPR and tryptophan catabolism were significantly upregulated at 8hpi during STM infection but not during SE and ST infection. In contrast, we found that cholesterol metabolism pathways were highly enriched during ST infection while amino acid metabolism, cellular responses to hypoxia, the inflammasome and antigen presentation pathways were only significantly induced in SE-infected HIOs. Although vesicular trafficking has been shown to play a critical role during *Salmonella* infection [16], only HIOs infected with STM and ST serovars significantly upregulated many of these pathways.

Next, we turned our attention to the downregulated DEG datasets. Consistent with our previous study [12], most of the significantly downregulated pathways during STM and ST infections belonged to cell cycle, DNA replication and repair, metabolism of protein and metabolism of RNA (**Fig 3A**), which point to potential reduction in cellular proliferation. Interestingly, in SE-infected HIOs, some of these categories including cell cycle and metabolism of proteins were instead associated with upregulated DEGs at 8hpi (**Fig 3A and 3B**). Taken together, we find that while most inflammatory responses are upregulated, other responses, including ECM, cellular stress, lipid and amino acid metabolism and cell cycle are differentially regulated upon infection with these three *Salmonella* serovars.

## *Salmonella* serovars induce distinct HIO proinflammatory response profiles

Intestinal epithelial cells initiate inflammatory responses via production of proinflammatory mediators. Because the most dramatic transcriptional responses we observed were related to immune signaling, we sought to identify the HIO signature of inflammatory mediators including chemokines, cytokines and antimicrobial peptides (AMP) in response to each *Salmonella* serovar (**Figs 4A–4C and S6**). We found that all these mediators were induced early during infection although with different magnitudes. For example, Colony Stimulating Factor 3 (CSF3), Interleukin 17C (IL17C), Interleukin 19 (IL19), C-C Motif Chemokine Ligand 20 (CCL20), C-X-C Motif Chemokine Ligand 1 (CXCL1), Defensin beta 4A (DEF4BA) and Peptidase Inhibitor 3 (PI3) were highly induced during STM infection, moderately induced during SE infection and only weakly induced during ST infection. ST is thought to evade detection from the immune system through expression of the Vi capsule [17]. Indeed, Vi capsule is induced in the static culture conditions prior to microinjection and was observed in HIOs during infection (**S7 Fig**). Of interest, IL17C signaling regulates epithelial host defense against mouse enteric pathogens [18]. HIO production of IL17C and its known downstream

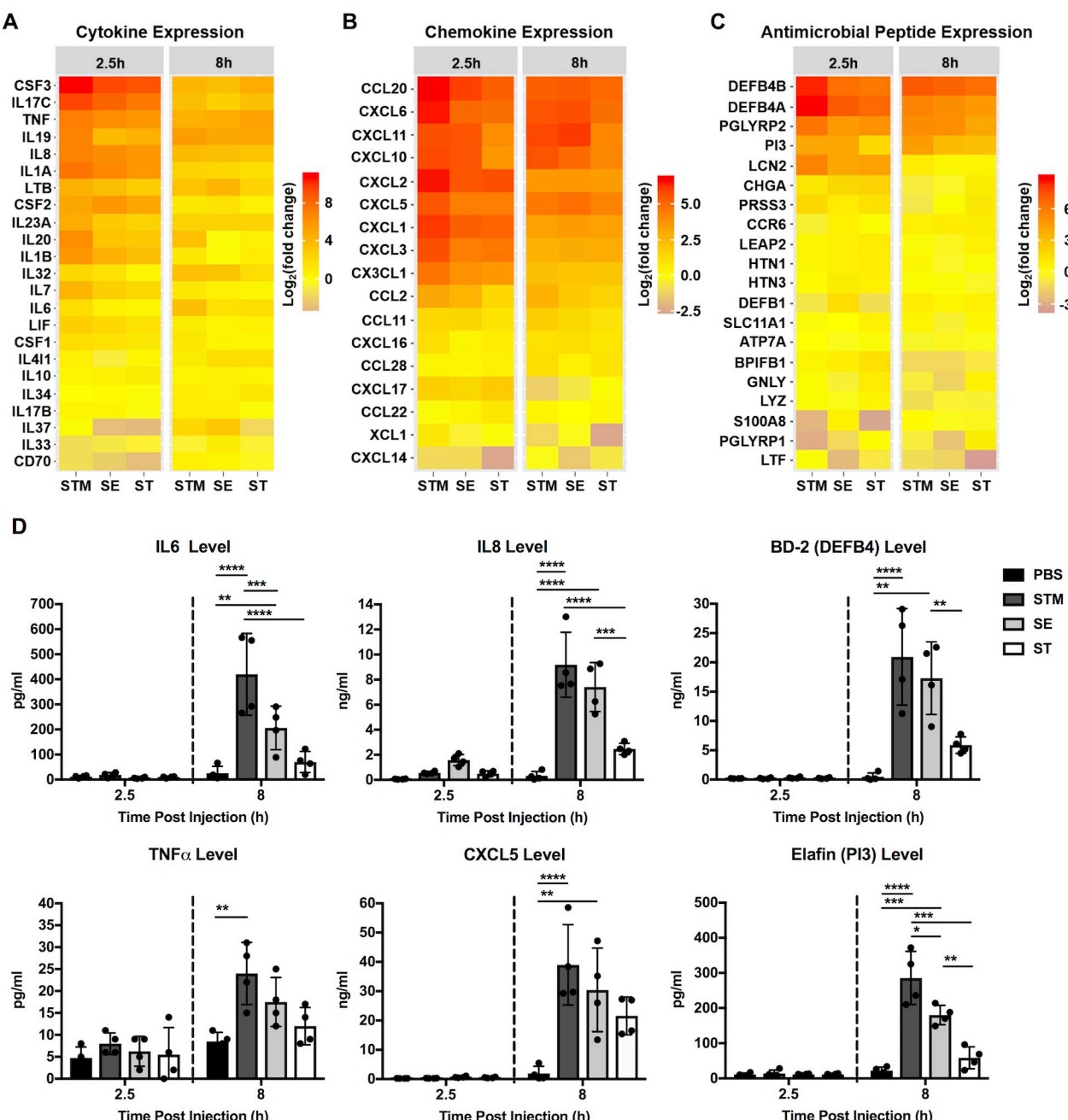

**Fig 4. Differential gene expression and secretion of immune modulators by HIOs in response to infection.** (A-C) Gene expression of Cytokine (A), Chemokine (B) and Antimicrobial peptide (C) are presented as log2 fold change relative to PBS at 2.5h and 8h post infection. (D) Cytokine, chemokine and antimicrobial peptide levels measured from HIO supernatant at 2.5h and 8h post infection via ELISA. n = 4 biological replicates. Error bars represent +/-SD. Significance calculated by two-way ANOVA. P values < 0.05 were considered significant and designated by: *<0.05, **<0.01, ***<0.001 and ****<0.0001.

proinflammatory mediators, including CSF3 and DEF4BA, also suggest that IL17C signaling modulates human intestinal host defense against *Salmonella* infection. To validate these transcriptional results, we measured production of specific inflammatory mediators (cytokine, chemokine and AMP) in the HIO culture medium by ELISA. All three serovars induced production of these inflammatory proteins (**Figs 4D and S8**). In general, changes in protein level

correlated with transcriptional changes observed in our RNA-seq dataset, where STM-infection resulted in the highest levels of cytokine production, SE-infection resulted in an intermediate phenotype and ST-infection induced the lowest levels. Collectively, the data indicate that each serovar, even the two non-typhoidal serovars, interacts distinctly with the host to tune production of inflammatory mediators during infection.

### Host cell cycle and cell death pathways are regulated during STM infection, but not during SE or ST infection

Our previous study and several others showed that STM infection decreases host cell proliferation [12,19,20]. To further investigate how cell cycle genes change in response to each serovar, we filtered the significant gene sets to examine expression patterns of cell cycle genes (Fig 5A). Although some downregulation occurred early during infection, a stronger effect was

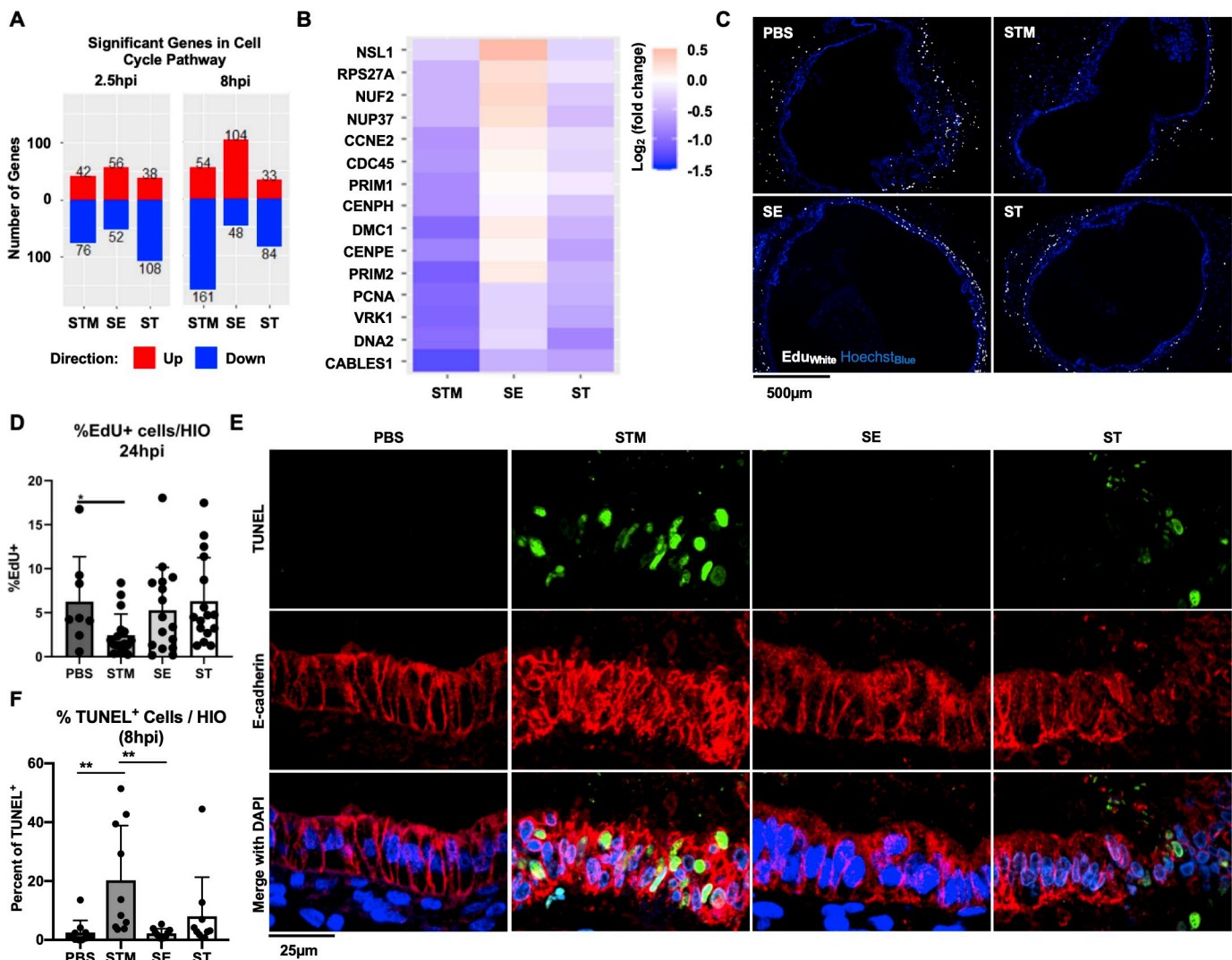

**Fig 5. STM suppresses host cell cycle and induces cell death.** (A) The number of significant genes in the cell cycle pathway upregulated (red) or downregulated (blue) in each condition. (B) Top differentially expressed cell cycle genes sorted by greatest standard deviation between STM and SE conditions. (C) EdU staining in HIOs at 24hpi. EdU (white), Hoechst (blue). (D) Quantitation of EdU staining with n>8 HIOs. (E) TUNEL staining in HIOs at 8hpi. TUNEL (green) E-cadherin (red) DAPI (blue). (F) Quantitation of TUNEL staining with n = 10 HIOs. Significance was determined by unpaired t-test where P value: * <0.05 and ** <0.01.

measured at 8hpi where most cell cycle-regulated genes were suppressed in response to both STM and ST infection, although notably there were fewer significant DEGs associated with ST infection. In contrast, during SE infection, most of the cell cycle DEGs were upregulated, suggesting that STM and ST may reduce HIO cell proliferation while SE infection may uniquely increase it. To better understand which genes were responding and how gene level expression patterns differed between serovars, a heatmap showing the genes most variable between infection conditions was generated (**Fig 5B**). Several genes critical for regulation of cell cycle progression were significantly downregulated during STM infection, including PCNA and VRK1 which normally increase in expression in dividing cells and PRIM1/2 encoding the two subunits of DNA primase important for initiating DNA replication during cell division [21–23]. These genes were observed to be downregulated, although weakly, during ST infection, and either weakly downregulated or upregulated during SE infection. To determine whether changes in cell cycle-related transcripts affected cell cycle progression, HIOs microinjected with PBS, STM, SE or ST were labeled with EdU to monitor proliferating cells for a period of 24h (**Fig 5C and 5D**). Consistent with the observation that STM suppressed expression of critical cell cycle genes, there was a significant reduction in the number of cells that incorporated EdU in STM-infected HIOs compared to PBS controls. In contrast, there was no significant change in the number of EdU+ cells in either ST or SE-infected HIOs suggesting that changes in gene expression that occurred during these infections were not sufficient to functionally alter cell cycle progression in the HIOs.

Cell cycle and cell death pathways are both involved in maintaining intestinal homeostasis during infection. Since cell death is known to play a strong role during *Salmonella* infection and was highly represented in our Reactome pathway analysis in STM-infected HIOs (**Fig 3A**), we measured cell death in the HIOs in response to all three serovars by performing Terminal deoxynucleotidyl transferase dUTP nick end labeling (TUNEL) (**Fig 5E and 5F**). Consistent with our pathway analysis, we found that STM infection resulted in a greater number of TUNEL-positive cells per HIO compared to PBS control. Surprisingly, the other NTS, SE, did not induce cell death at greater frequency compared to controls suggesting that the two NTS interact quite differently with the host. In contrast, ST induced an intermediate response with some HIOs exhibiting increased host cell death compared to the PBS control. Together these results from the HIO model show that STM disrupts intestinal epithelial homeostasis to a greater degree than SE or ST, at least in part by suppressing host cell division and inducing host cell death.

## Mitochondrial processes are differentially regulated during NTS infections

Although NTS cause similar disease manifestations in humans, they may interact with the intestinal epithelium by varied mechanisms as their genomes contain different accessory genes [24]. Our data indicated that one of the most differentially regulated cellular processes between NTS was related to metabolism of proteins (**Fig 3A**). To further identify major pathways within this category that were differentially regulated during infection with NTS, we sorted significant pathways that belonged to the metabolism of proteins category in the Reactome database to identify these pathways. We found pathways belonging to three major categories; translation, protein folding, and post-transcriptional regulation were increased in SE-infected HIOs but decreased during STM infection (**Fig 6A**). Within the translation umbrella category, we found many mitochondrial-related processes, including mitochondrial translation, mitochondrial protein import and oxidative phosphorylation, were increased during SE infection but decreased during STM infection (**Fig 6B**), suggesting that mitochondrial functions may differentiate between the host response to NTS during early stages of infection. Because

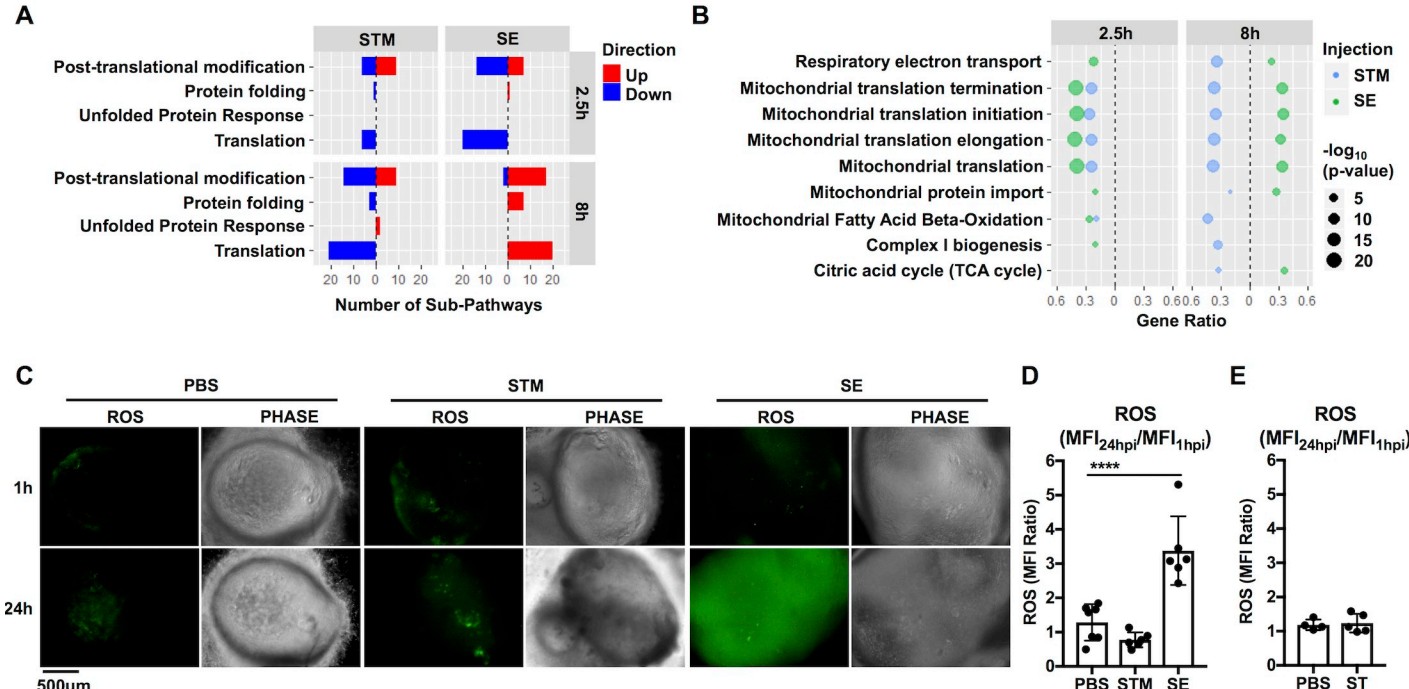

**Fig 6. NTS infections inversely regulate changes in mitochondrial-related cellular processes at 8hpi and trigger differential ROS production.** (A) Number of significant sub-pathways from the Reactome metabolism of proteins category during NTS infection. Upregulated (red) or downregulated (blue) pathways were identified using ReactomePA. Significant pathways were determined based on P value < 0.05. (B) Dot plot showing significantly enriched mitochondrial metabolism-related pathways in NTS-infected HIOs. Filtered by P value < 0.05. (C) Representative fluorescence images measuring reactive oxygen species (ROS) levels in HIOs by general oxidative stress dye, CM-H$_2$DCFDA. (D) Quantitation reactive oxygen species levels of (C) with n≥6 HIOs measuring the ratio of the mean fluorescence intensity (MFI Ratio) at 24h relative to 1hpi. (E) Quantification of reactive oxygen species levels in PBS or ST-infected HIOs at 24h relative to 1h. Error bars represent +/-SD. Statistical significance was determined by one-way ANOVA followed up by Tukey's multiple comparisons test. ****<0.0001.

mitochondria produce reactive oxygen species (ROS) during metabolism, we monitored generation of ROS in HIOs during infection (**Fig 6C–6E**). Consistent with an increase in mitochondrial gene expression, SE infection led to an accumulation of ROS in the HIOs when compared to PBS control. ROS induction was specific to SE, as neither STM nor ST infection triggered ROS generation when monitored at 24hpi. Together, our results suggest that SE infection induces specific HIO responses, including induction of mitochondria-related processes and ROS generation that distinguish this serovar from the more well-studied STM serovar.

## Discussion

Despite sharing high genome identity, some *Salmonella* serovars cause infections that remain localized in the intestine while others cause more severe systemic infections. Differential host responses, especially the initial interactions with the human intestinal epithelium, are likely a contributing factor in determining infection outcome, something that has been difficult to study with other established infection models. Here we describe the first comprehensive transcriptomic analysis using human intestinal organoids infected with *Salmonella enterica* serovars Typhimurium, Enteritidis and Typhi. We compared HIO transcriptional profiles at different time points during infection and identified patterns that were similar and unique to each serovar. As expected, inflammatory pathways dominated early responses to infection with all three serovars. However, at later times post infection, we observed distinct

transcriptomes associated with each serovar with differences in expression of genes in cell cycle and mitochondrial function-related pathways. Direct comparison of HIO responses to these serovars revealed that many pathways decreased during STM infection were, in contrast, increased during SE infection even though these serovars cause similar diseases in humans. Thus, our data highlight the utility of the HIO model to define signatures of host responses to closely related enteric pathogens to understand how this early programming may shape disease manifestations.

ST is a human-specific pathogen and other serovars exhibit different infection patterns in different organisms [25,26], therefore it is physiologically relevant to use human-derived cells to define human mechanisms of pathogenesis and host response. Many previous studies investigating host epithelial responses employed transformed cell lines [13,27–30], and although they have provided valuable insight into the specific mechanisms of *Salmonella* pathogenesis, particular aspects of epithelial function may not fully reflect what happens in an untransformed model system [31]. More in-depth comparisons of host epithelial responses using different culture models revealed that approximately 50% of the top significantly regulated genes reported in a study published by Hannemann *et al.* [13] investigating responses to STM in a Henle-407 cell culture model were significant in our study while a study using a stem cell derived intestinal organoid culture model, published by Forbester *et al.* [11], showed a much higher correlation of epithelial gene expression to our dataset with about 90% of the reported top significant genes also being significant in our study. In a more complex model system (an engineered organotypic model) [14], the correlation between datasets was relatively low with only about 30% similarity suggesting that addition of other components such as immune cells strongly contributes to the host response to infection. Additionally, studies looking at host responses to ST showed largely the same pattern [13,27,28,32]. Across the various studies, the conserved responses largely belonged to immune-related processes suggesting that other types of responses may be more dependent on the specific model system. Our study aimed to compare and contrast three major *Salmonella enterica* serovars that are human pathogens, and we chose commonly used laboratory strains to represent each serovar. It is important to note that there may be meaningful differences in the epithelial host response even between strains within the same serovar. Although these findings merit a more detailed analysis to better understand how different model systems respond to each serovar or to different strains within serovars, we reason that the HIOs, composed of untransformed human intestinal epithelial and mesenchymal cells, might reveal specific responses that more closely mirror *in vivo* epithelial responses to infection.

Comparing different parameters of HIO infections revealed marked differences between the three serovars. Of particular interest was the efficiency with which STM infected the epithelium when the HIOs were injected with a relatively low inoculum ($10^3$), compared to SE and ST despite similar overall bacterial numbers within the HIO. This phenomenon could be explained by the earlier expression of genes encoding some SPI-1-dependent effectors in STM, especially SopE. In contrast, of the three serovars, SE expressed SPI-2 related genes most robustly at 8hpi, which correlated with a substantial increase in intracellular bacteria. These variations between serovars in early interactions with the host may direct the timing and magnitude of some aspects of the innate immune response, based on relative proportions of luminal to intracellular bacteria. Additionally, intracellular STM can be found in either the vacuole or the cytosol, and a recent study identified specific bacterial adaptations required for these two different intracellular lifestyles [33]. In the HIOs, we observed that some inflammatory pathways were decreased early during ST infection when compared to NTS. However, this effect was abrogated by 8hpi, emphasizing that ST may specifically modulate intestinal epithelial responses early and transiently during infection. Infection with each serovar stimulated

distinct transcriptional profiles of numerous inflammatory mediators, potentially contributing to the difference in serovar-specific pathogenicity. We demonstrated that ST infection produced less secreted IL-6, IL-8, BD-2 and ELAFIN (PI3) compared to STM and SE infection, despite comparable increases in transcript levels. These findings lead us to speculate that ST may impair the ability of intestinal epithelial cells to release immune mediators through secretion blockade or post-transcriptional modification. Consistent with the hypothesis that ST disrupts secretion pathways in epithelial cells, we also observed accumulation of mucus within epithelial cells of ST-infected HIOs in contrast to STM-infected HIOs where mucus was expelled into the luminal space. Post-transcriptional control of cytokine production has also been previously established, but not in the context of *Salmonella* infection [34–36]. Whether these mechanisms control ST pathogenesis and intestinal inflammatory responses are unknown, but could be elucidated in the HIO model.

Our finding that the three *Salmonella* serovars showed differential regulation of cell cycle pathways was intriguing. Intestinal epithelial cells undergo self-renewal to maintain barrier integrity, and infection with enteric pathogens can accelerate or inhibit cell proliferation to gain a survival advantage in the gut [37]. For example, *Citrobacter rodentium* stimulates the proliferation of undifferentiated epithelial cells, which increases oxygenation of the mucosal surface in the colon to create a replicative niche [38]. By contrast, some enteric pathogens, including STM, *Helicobacter pylori* and *Shigella* species are equipped with virulence factors to counteract intestinal cell proliferation and rapid epithelial turnover to enhance virulence [37]. In our experiments, both STM and ST infections resulted in downregulation of many genes in the cell cycle pathway while SE infection resulted in upregulation of several of these genes. Follow-up studies measuring cellular proliferation revealed that STM infection resulted in a reduction of proliferating cells in the HIO, but no change was observed in SE or ST-infected HIOs. Of note, it was previously reported that STM blocks epithelial cell proliferation via Type 3 Secretion System-2 effectors SseF and SseG [19]. These effectors are also encoded in the ST and SE genomes and were expressed in the HIOs at higher levels than in STM infection, suggesting that there may be alternative mechanisms leading to cell cycle suppression that remain to be uncovered.

Although STM and SE cause similar diseases in humans, we were surprised to observe that these two serovars exhibited the most variation in HIO responses relative to each other, including regulation of mitochondrial function-related genes. We previously showed that mitochondrial ROS plays a key role in shaping innate immune responses to bacterial infection and contributes to bacterial killing by macrophages [39]. Interestingly, we observed that many pathways involved in mitochondrial metabolism are upregulated during SE infection and downregulated during STM infection. Accordingly, we found that SE infection increased generation of antimicrobial ROS in the HIOs, suggesting that an increase in mitochondrial metabolism may be important in intestinal host defense. Indeed, mitochondrial integrity and function is required for the maintenance of healthy intestinal barriers to prevent bacterial translocation across the epithelial lining [40,41]. In addition, recent studies demonstrated that metabolites produced by microbes in the gut can influence mitochondrial biogenesis and inflammation [42]. Given that both STM and SE are present in the HIO lumen through the course of infection, it remains unclear whether SE uniquely increases expression of mitochondrial genes, or luminal bacteria generally increase expression of mitochondrial genes but STM uniquely decreases their expression, or both. SE encodes more than 200 genes that are absent in either the STM or ST genome, which are clustered in unique islands designated as "regions of difference" (ROD) [5]. Some of these additional genes have been linked to SE pathogenesis using a mouse model of *Salmonella* infection [43,44]. Therefore, we speculate that genes

expressed only by SE might account for SE-specific induction of mitochondrial ROS and further work is required to elucidate mechanisms by which SE induces these specific responses.

Altogether, our findings show that HIOs are a productive model to study early interactions of *Salmonella* serovars with the intestinal epithelium. HIOs have been previously used to probe for transcriptional responses during STM infection [11,12], but to our knowledge this is the first study to directly compare non-transformed human intestinal epithelial responses between non-typhoidal and typhoidal serovars. Looking beyond the pro-inflammatory pathways induced during infection by all three serovars, we identified unique host responses that are individually associated with these closely related serovars. Patterns emerging from our HIO experiments open up avenues for future studies to elucidate mechanisms by which different serovars fine-tune inflammatory output and modulate cell cycle and mitochondrial functions.

# Materials and methods

## HIO differentiation and culture

HIOs were generated by the *In Vivo* Animal and Human Studies Core at the University of Michigan Center for Gastrointestinal Research as previously described [45]. Briefly, human ES cell line WA09 was obtained from Wicell International Stem Cell Bank and cultured on Matrigel-coated (BD Biosciences) 6-well plates in mTeSR1 media (Stem Cell Technologies) at 37˚C in 5% $CO_2$. Cells were seeded onto Matrigel-coated 24-well plates in fresh mTeSR1 media and grown until 85–90% confluence. Definitive endoderm differentiation was induced by washing the cells with PBS and culturing in endoderm differentiation media (RPMI 1640, 2% FBS, 2mM L-glutamine, 100ng/ml Activin A, 100U/ml of Penicillin and 100μg/ml of Streptomycin) for three days where fresh medium was added each day. Cells were then washed with endoderm differentiation media without Activin A and cultured in mid/hindgut differentiation media (RPMI 1640, 2% FBS, 2mM L-glutamine, 500ng/ml FGF4, 500ng/ml WNT3A, 100U/ml of Penicillin and 100μg/ml of Streptomycin) for 4days until spheroids were present. Spheroids were collected, mixed with ice cold Matrigel (50 spheroids + 50μl of Matrigel + 25μl of media), placed in the center of each well of a 24-well plate, and incubated at 37˚C for 10min to allow Matrigel to solidify. Matrigel embedded spheroids were grown in ENR media (DMEM:F12, 1X B27 supplement, 2mM L-glutamine, 100ng/ml EGF, 100ng/ml Noggin, 500ng/ml Rspondin1, and 15mM HEPES) for 14 days where media were replaced every 4days. Spheroids growing into organoids (HIOs) were dissociated from the Matrigel by pipetting using a cut wide-tip (2-3mm). HIOs were mixed with Matrigel (6 HIOs + 25μl of media + 50μl of Matrigel) and placed in the center of each well of 24-well plates and incubated at 37˚C for 10min. HIOs were further grown for 14days in ENR media with fresh media every 4days. Prior to experiments, HIOs were carved out of the Matrigel, washed with DMEM:F12 media, and re-plated with 5 HIO/well in 50μl of Matrigel in ENR media with media exchanged every 2–3 days for 7days prior to microinjection.

## Bacterial growth conditions and HIO microinjection

*Salmonella* strains used in this study are *Salmonella* enterica serovar Typhimurium strain SL1344, *Salmonella* enterica serovar Enteritidis strain P125109 and *Salmonella* enterica serovar Typhi strain Ty2. Strains were stored at -70˚C in LB medium containing 20% glycerol and cultured on Luria-Bertani (LB, Fisher) agar plates. Selected colonies were grown overnight at 37˚C under static conditions in LB liquid broth. Bacteria were pelleted, washed and re-suspended in PBS. The bacterial inoculum was estimated based on $OD_{600}$ and verified by plating serial dilutions on agar plates to determine colony forming units (CFU). HIOs were cultured

in group of 5 per well using 4-well plates (Thermo Fisher). Lumens of individual HIOs were microinjected with glass caliber needles with 1μl of PBS or different strains of *Salmonella* ($10^5$ CFU/HIO for 2.5h or 8h for RNAseq experiments or $10^3$ CFU/HIO for 2.5h or 24h for bacterial burden experiments). HIOs were then washed with PBS and incubated for 2h at 37˚C in ENR media. After 2h, HIOs were treated with 100μg/ml gentamicin for 15 min to kill any bacteria outside the HIOs, then incubated in a fresh medium containing 10μg/ml gentamicin.

## ELISA and bacterial burden analyses

Media from each well (5 HIOs/well) were collected at indicated time points after microinjection. Cytokines, chemokines and defensins were quantified by ELISA at the University of Michigan Cancer Center Immunology Core. Bacterial burden was assessed per HIO. Individual HIOs were removed from the Matrigel, washed with PBS and homogenized in PBS. Total CFU/HIO were enumerated by serial dilution and plating on LB agar. To assess intracellular bacterial burden, HIOs were cut open, treated with 100μg/ml gentamicin for 10min to kill luminal bacteria, washed with PBS, homogenized and plated on agar plates for 24h.

## Immunohistochemistry and immunofluorescence staining

HIOs were fixed with either 10% neutral formalin or Carnoy's solution for 2 days and embedded in paraffin. Histology sections (5μm) were collected by the University of Michigan Cancer Center Histology Core and stained with hematoxylin and eosin (H&E). Carnoy's-fixed HIO sections were stained with Alcian Blue and Periodic Acid-Schiff (PAS) staining kit according to the manufacturer's instructions (Newcomersupply). H&E and Alcian Blue/PAS-stained slides were imaged on an Olympus BX60 upright microscope. All images were further processed using ImageJ. For immunofluorescence staining, formalin-fixed HIO sections were deparaffinized prior to performing antigen retrieval in sodium citrate buffer (10mM Sodium citrate, 0.05% Tween 20, pH 6.0). Sections were permeabilized with PBS + 0.2% Triton X-100 for 30min, then incubated in a blocking buffer (PBS, 5% BSA) for 1h. E-cadherin was stained using mouse anti-E-cadherin polyclonal antibody (clone 36, BD Biosciences) in a blocking buffer overnight at 4˚C. Goat anti-mouse secondary antibody conjugated to Alexa-594 was used according to manufacturer's instructions (Thermo Fisher) for 1h RT in blocking buffer. To measure cell death, HIOs were stained using the *In situ* Cell Death Detection kit (Roche) following the manufacturer's instructions for paraffin fixed tissue. DAPI was used to stain DNA. Sections were mounted using coverslips (#1.5) and Prolong Glass Antifade Mountant (Thermo Fisher). Images were taken on a Nikon X1 Yokogawa spinning disc confocal microscope and processed using ImageJ and CellProfiler.

## Vi capsule detection

Vi expression was monitored by flow cytometry. Bacteria grown under static or aerated conditions were washed with PBS and stained with *Salmonella* Vi Rabbit Antiserum (BD Biosciences) using 1:100 dilution followed by staining with goat anti-rabbit secondary antibody conjugated to PE. Bacteria were analyzed on FACSCanto Cell Analyzer (BD Biosciences). Data were further analyzed with FlowJo software and percent of Vi-positive bacteria was determined by gating against unstained cells. For histology staining, immunofluorescence was performed on 5μm histology sections of ST-infected HIOs using *Salmonella* Vi Rabbit Antiserum (1:250 dilution, BD Biosciences) followed by a secondary goat anti-rabbit antibody conjugated to Alexa 488 (Thermo Fisher). DAPI was used to stain DNA. Images were taken on an Olympus BX60 upright microscope. All fluorescence images were processed and analyzed using ImageJ.

## Cell proliferation analysis

After microinjection, 25μM EdU was added to the HIO culture medium and incubated at 37°C for 24h to allow incorporation into dividing cells. HIOs were fixed and sectioned as outlined above and stained using the Click-iT EdU Cellular Proliferation kit (Thermo Fisher) according to the manufacturer's protocol. HIOs were counterstained with Hoechst to detect DNA before mounting in Prolong Glass Antifade Mountant (Thermo Fisher). Images were taken on an Olympus BX60 upright microscope and processed and analyzed using ImageJ and CellProfiler.

## RNA sequencing and analysis

Total RNA was isolated from groups of 5 HIOs per replicate with a total of 4 replicates per infection condition using the mirVana miRNA Isolation Kit (Thermo Fisher). The data shown here are part of a larger sample set that were analyzed for multiple studies. Thus, RNAseq data from the PBS and STM samples are included in a previously published study [12], however, the SE and ST sample data and the associated analyses are unique to this study. The quality of RNA was confirmed, RNA integrity number (RIN)>8.5, using the Agilent TapeStation system. cDNA libraries were prepared by the University of Michigan DNA Sequencing Core after cytosolic and mitochondrial ribosomal RNA depletion from samples using the TruSeq Stranded Total RNA with Ribo-Zero Gold Kit according to the manufacturer's protocol (Illumina). Libraries were sequenced on Illumina HiSeq 2500 platforms (single-end, 50bp read length). All samples were sequenced at a depth of 12 million reads per sample or greater.

## Bioinformatics comparison with previously published transcriptomics studies

Gene lists of the top 30–50 genes from indicated publications were used to filter against significant gene changes in infected HIOs with the corresponding serovar at either 2.5h or 8hpi. The fraction of genes from each list that were significant in the HIO dataset was then calculated. Log$_2$(fold change) of these significant genes was also determined and plotted using ggplot2 in R [46].

## RNA-seq analysis protocol

**Sequence alignment.** Sequencing generated FASTQ files of transcript reads were pseudoaligned to the human genome (GRCh38.p12) using kallisto software [47]. Transcripts were converted to estimated gene counts using the tximport package with gene annotation from Ensembl [48,49].

**Differential gene expression.** Differential expression analysis was performed relative to PBS samples at each time point using the DESeq2 package with P values calculated by the Wald test and adjusted P values calculated using the Benjamani & Hochberg method [50,51].

**Pathway enrichment analysis.** Pathway analysis was performed using the Reactome pathway database and pathway enrichment analysis in R using the ReactomePA software package [52].

**RNAseq statistical analysis.** Analysis was done using RStudio version 1.1.453. Plots were generated using ggplot2 with data manipulation done using dplyr [46,53]. Euler diagrams of gene changes were generated using the Eulerr package [54].

**Reactive oxygen species (ROS) measurement.** HIOs were re-plated onto glass-bottom petri dishes (MatTek) and microinjected with 1μl of PBS or bacteria containing 50ng of

CM-H$_2$DCFDA per HIO (Thermo Fisher). HIOs were imaged using inverted widefield live fluorescent microscopy at indicated time points. Images were analyzed by ImageJ.

## Quantification and statistical methods

Data were analyzed using Graphpad Prism 7 and R software. Source data can be found in **S1 Data**. Statistical differences were determined using one-way ANOVA or two-way ANOVA (for grouped analyses) and followed-up by Tukey's multiple comparisons test. The mean of at least 3 independent experiments were presented with error bars showing standard deviation (SD). P values of less than 0.05 were considered significant and designated by: *<0.05, **<0.01, ***<0.001 and ****<0.0001.

## Supporting information

**S1 Fig. Temporal regulation of SPI-1 and SPI-2 gene expression in the HIOs by different *Salmonella* serovars.** (A and B) SPI-1 and SPI-2 gene expression normalized to RecA expression at each time point in the HIOs. Log$_2$(fold change) at 8h relative to 2.5hpi was calculated and shown in the bottom row.
(TIFF)

**S2 Fig. Quantification of Alcian blue and periodic acid-Schiff (PAS) staining.** (A) Luminal and intracellular staining intensity from n>4 HIOs based on images shown in Fig 1G. Significance was determined by one-way ANOVA where P value: *<0.05 and ****<0.0001.
(TIFF)

**S3 Fig. Similar bacterial loads are present in *Salmonella* serovar-infected HIOs.** HIOs were infected with 10$^5$ CFU of STM, SE or SE and total bacterial burden per HIO was enumerated at 8hpi. Graph represents the mean of n>8 HIOs.
(TIF)

**S4 Fig. Comparison of HIO responses with previously published transcriptomics studies investigating host cell responses to *Salmonella* infection.** Top 30–50 genes reported in publications listed above were compared to our significant gene sets. The percentage of those genes that were also significant in either STM or ST-infected HIOs was plotted in the bar graphs with the model system used in each study listed at the top of each bar. Conserved gene changes were plotted in heatmaps to compare fold change across the different model systems.
(TIFF)

**S5 Fig. Select Reactome pathways that are differentially upregulated during HIO infection with different *Salmonella* serovars are related to antigen presentation, extracellular matrix, cellular stress responses, vesicular trafficking, lipid metabolism, and amino acid metabolism.** Dotplot shows select Reactome pathways that are significantly enriched (P value < 0.05) from upregulated gene sets of HIOs infected with different *Salmonella* serovars relative to PBS control.
(TIFF)

**S6 Fig. Complete cytokine gene list from Reactome.** Gene expression presented as log$_2$ fold change during STM, SE or ST infection relative to PBS at 2.5h and 8hpi.
(TIFF)

**S7 Fig. *Salmonella* typhi strain Ty2 expresses the Vi polysaccharide capsule when cultured *in vitro* under static conditions and in the HIO.** (A) Representative flow cytometry histograms of Vi polysaccharide capsule expression by STM and ST. (B) Vi polysaccharide capsule

expression was quantified by flow cytometry using rabbit Vi antisera. STM strain SL1344 and ST strain Ty2 were cultured overnight at 37°C under static or shaking conditions. Bacteria were washed, stained with rabbit Vi antisera and analyzed by flow cytometry. Percent capsule[+] cells was determined by gating against unstained cells. Graph indicates means +/- SD of n≥3 experiments. (C) Representative fluorescence microscopy images of ST-infected HIOs at 8hpi. Sections were stained with Rabbit antisera (green), anti-E Cadherin antibody (red) and DAPI (blue). P-value was calculated using two-way ANOVA with Sidek's post-test for multiple comparisons. P value: ** <0.01 and **** <0.0001.
(TIFF)

**S8 Fig. Chemokine secretion levels at 2.5h and 8hpi for HIOs microinjected with PBS, STM, SE, and ST.** Graphs are presented as mean of n = 4 biological replicates with standard deviation (SD) error bars. P value was calculated using two-way ANOVA with Tukey's post-test for multiple comparisons. P value: * <0.05; ** <0.01, *** <0.001 and **** <0.0001.
(TIFF)

**S1 Table. DEGs of STM, SE and ST-infected HIOs at 2.5h relative to PBS control.** Significant DEGs (P < 0.05) in at least one infection condition are listed.
(PDF)

**S2 Table. DEGs of STM, SE and ST-infected HIOs at 8h relative to PBS control.** Significant DEGs (P < 0.05) in at least one infection condition are listed.
(PDF)

**S3 Table. Enriched Reactome pathways from upregulated DEGs at 2.5hpi.** Significantly upregulated DEGs from STM, SE or ST-infected HIOs at 2.5h were subjected to Reactome pathway analysis.
(PDF)

**S4 Table. Enriched Reactome pathways from downregulated DEGs at 2.5hpi.** Significantly downregulated DEGs from STM, SE or ST-infected HIOs at 2.5h were subjected to Reactome pathway analysis.
(PDF)

**S5 Table. Enriched Reactome pathways from upregulated DEGs at 8hpi.** Significantly upregulated DEGs from STM, SE or ST-infected HIOs at 8hpi were subjected to Reactome pathway analysis.
(PDF)

**S6 Table. Enriched Reactome pathways from downregulated DEGs at 8hpi.** Significantly downregulated DEGs from STM, SE or ST-infected HIOs at 8hpi were subjected to Reactome pathway analysis.
(PDF)

**S1 Data. Source data used in main and supplemental figures.**
(XLSX)

## Acknowledgments

We thank the Host Microbiome Initiative, the Center for Live Cell Imaging (CLCI), Microscopy and Image Analysis Laboratory (MIL), the Comprehensive Cancer Center Immunology and Histology Cores and the DNA Sequencing Core at the University of Michigan Medical School. We thank C. Detweiler (Univ. of CO, Boulder) and H. Andrews-Polymenis (Texas

A&M) for the kind gift of bacterial strains. We gratefully acknowledge the O'Riordan lab members for helpful discussions.

## Author Contributions

**Conceptualization:** Basel H. Abuaita, Christiane E. Wobus, Jason R. Spence, Vincent B. Young, Mary X. O'Riordan.

**Data curation:** Basel H. Abuaita, Anna-Lisa E. Lawrence, Ryan P. Berger, Mary X. O'Riordan.

**Formal analysis:** Basel H. Abuaita, Anna-Lisa E. Lawrence, Ryan P. Berger.

**Funding acquisition:** Christiane E. Wobus, Jason R. Spence, Vincent B. Young, Mary X. O'Riordan.

**Investigation:** Basel H. Abuaita, Anna-Lisa E. Lawrence, Ryan P. Berger, Mary X. O'Riordan.

**Methodology:** Basel H. Abuaita, Anna-Lisa E. Lawrence, Ryan P. Berger, David R. Hill, Sha Huang, Veda K. Yadagiri, Brooke Bons, Courtney Fields, Jason R. Spence, Vincent B. Young.

**Project administration:** Christiane E. Wobus, Jason R. Spence, Vincent B. Young, Mary X. O'Riordan.

**Resources:** David R. Hill, Sha Huang, Veda K. Yadagiri, Brooke Bons, Courtney Fields, Jason R. Spence, Mary X. O'Riordan.

**Software:** Anna-Lisa E. Lawrence, Ryan P. Berger, David R. Hill.

**Supervision:** Basel H. Abuaita, Mary X. O'Riordan.

**Validation:** Basel H. Abuaita, Anna-Lisa E. Lawrence, Mary X. O'Riordan.

**Visualization:** Basel H. Abuaita, Anna-Lisa E. Lawrence, Mary X. O'Riordan.

**Writing – original draft:** Basel H. Abuaita, Anna-Lisa E. Lawrence, Ryan P. Berger, Mary X. O'Riordan.

**Writing – review & editing:** Basel H. Abuaita, Anna-Lisa E. Lawrence, Christiane E. Wobus, Jason R. Spence, Vincent B. Young, Mary X. O'Riordan.

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
