## [Decision Letter · Decision Letter 0]

5 Aug 2021

Dear Professor O'Riordan,

Thank you very much for submitting your manuscript "Comparative transcriptional profiling of the early host response to infection by typhoidal and non-typhoidal Salmonella serovars in human intestinal organoids" for consideration at PLOS Pathogens. As with all papers reviewed by the journal, your manuscript was reviewed by members of the editorial board and by several independent reviewers, please accept our apologies for the exceptionally long process in this instance. The reviewers appreciated the attention to an important topic. Based on the reviews, we are likely to accept this manuscript for publication, providing that you modify the text of the manuscript according to the review recommendations.

Sincerely,

Sophie Helaine

Associate Editor

PLOS Pathogens

Nina Salama

Section Editor

PLOS Pathogens

Kasturi Haldar

Editor-in-Chief

PLOS Pathogens

orcid.org/0000-0001-5065-158X

Michael Malim

Editor-in-Chief

PLOS Pathogens

orcid.org/0000-0002-7699-2064

Reviewer Comments (if any, and for reference):

Reviewer's Responses to Questions

**Part I - Summary**

Reviewer #1: In the revised version of this study, Abuaita and colleagues have successfully addressed a majority of the comments and have provided sound experimental evidence to validate the transcriptome analysis. While the manuscript remains largely descriptive, it now offers a solid and informative experimental base for a variety of interesting follow-up studies. Especially the newly added data on SPI1- and SPI2-expression by the different strains, as well as the comparison of expression signatures in different models and studies adds very interesting new insights and further validates HIOs as a model for studying host-microbe interaction. A few points however remain to be resolved.

Reviewer #2: Abuaita et al. provide an extensive transcriptomic characterization of a human intestinal organoids exposed to three Salmonella serovars (Typhimurium (STM), Enteritidis (SE) and Typhi (ST)). The originality of the paper lies on the use of the organoids that best mimic the in vivo conditions. I commend this paper because it constitutes an important resource for the field. The revision has clarified my concerns.

**Part II – Major Issues: Key Experiments Required for Acceptance**

Reviewer #1: - The authors have added some clarification regarding the inoculum sizes used, but it remains unclear which inoculum size was used for the 2.5h time point. It seems that in Fig. 1, 10^3 CFU were injected (according to l. 124-125), while for the transcriptome data set it sounds like the incoculum size for the 2.5h time point was 10^5 CFU (l. 173-175). Unfortunately, nothing is stated in the methods section regarding this time point. This point needs to be clarified, especially as the low inoculum seems to result in differences in Salmonella loads for the different strains tested, while there is no evident difference with the higher inoculum at 8 hpi. If the 2.5h infection for the transcriptome data set was indeed performed with a different inoculum size than the data presented in Fig. 1B, it would be highly important for the interpretability of the data set to add the CFU data also for this time point and inoculum size. Stating the respective inoculum sizes for the different experiments in the figure legends (Fig. 1 and 2) would make it much easier for the reader to understand the experimental setup.

- While it is evident that the authors cannot test numerous strains per serovar and it is reasonable to choose the best-characterized ones for the experimental setup used here, the substantial differences between strains of one serovar should anyhow be discussed to make clear to the reader that host responses likely vary substantially even between strains of one serovar. It should bepointed out that the response of one strain cannot be generalized to the entire serovar.

Reviewer #2: However, I still have a strong concern on the massive experimental variability between the three serovars that the authors outline in Figure 1 (Figure 1B and 1C). This variability is clearly bounded to the experimental set-up and I believe that further studies will refine this problem. Furthermore, in the current setting, the authors cannot discriminate the host response induced solely by the extracellular versus the intracellular bacteria. The discussion should clearly outline that the heterogeneity in the host response can be due to these confounding factors.

**Part III – Minor Issues: Editorial and Data Presentation Modifications**

Reviewer #1: - Please plot individual data points instead/in addition for Fig 5D, S3

- L. 103: "…to compare the transcriptomes of intestinal…"

- L. 174: please clarify what you mean with "similar" (similar to which condition?)

- L. 213: add "at 8 hpi" to clarify

- L. 319: "..during these infections were not sufficient…"

- L. 456-457 "…other serovars exhibit different infection patterns in different organisms": please provide references

- L. 487: "…despite inducing their…"

Reviewer #2: No Minor Issues

PLOS authors have the option to publish the peer review history of their article (what does this mean?). If published, this will include your full peer review and any attached files.

Reviewer #1: No

Reviewer #2: No

Figure Files:

Data Requirements:

Reproducibility:

References:

---

## [Editor Report · Decision Letter 1]

28 Sep 2021

Dear Prof O'Riordan,

We are pleased to inform you that your manuscript 'Comparative transcriptional profiling of the early host response to infection by typhoidal and non-typhoidal Salmonella serovars in human intestinal organoids' has been provisionally accepted for publication in PLOS Pathogens.

Best regards,

Sophie Helaine

Associate Editor

PLOS Pathogens

Nina Salama

Section Editor

PLOS Pathogens

Kasturi Haldar

Editor-in-Chief

PLOS Pathogens

orcid.org/0000-0001-5065-158X

Michael Malim

Editor-in-Chief

PLOS Pathogens

orcid.org/0000-0002-7699-2064
---

## [Editor Report · Acceptance letter]

18 Oct 2021

Dear Prof. O'Riordan,

We are delighted to inform you that your manuscript, "Comparative transcriptional profiling of the early host response to infection by typhoidal and non-typhoidal Salmonella serovars in human intestinal organoids," has been formally accepted for publication in PLOS Pathogens.

Best regards,

Kasturi Haldar

Editor-in-Chief

PLOS Pathogens

orcid.org/0000-0001-5065-158X

Michael Malim

Editor-in-Chief

PLOS Pathogens

orcid.org/0000-0002-7699-2064